# Cylindrical Planetary Nebulae. I. Flow from an Irradiated Ring

Vincent Icke

Sterrewacht Leiden, Universiteit Leiden, 2300 RA Leiden, The Netherlands; icke@strw.leidenuniv.nl

**Abstract:** Many bipolar nebulae with a pronounced cylindrical shape, such as Henize 3-401, show no indication whatsoever of interaction between a disk and a stellar wind, or a jet on the nebular axis. I propose that the disk that is observed at the base of the bipolar is itself the source of the outflow. In particular, I assume that irradiation from the central star causes the disk to evaporate. I have performed numerical hydrodynamical calculations of outflows driven by evaporation of a pseudo-barotropic ring around a hot central star. The first results show that the outflow shapes are cylindrical, and the internal structures are similar to what is observed in some of these nebulae. Since shape is only the first step in the assessment of a model, synthetic observations should be made. For the moment I merely verify that the scalar quantities observed in the archetypical cylindrical nebula Hen 3-401 can be accommodated in my models.

**Keywords:** bipolar nebulae; protoplanetary nebulae; Post-AGB stars; numerical hydrodynamics

## 1. Introduction

Most (pre)planetary nebulae are very aspherical, and many have a distinct bipolar shape. In the past decades, many authors have proposed mechanisms for the formation of such nebulae: interaction between a stellar wind and a circumstellar disk, supersonic jets emanating from a compact disk, and magnetic fields of various configurations. However, some bipolar nebulae with a pronounced cylindrical shape [1], such as in Figure 1, show no indication whatsoever of interaction between a disk and a stellar wind, or a jet on the nebular axis. Moreover, in these cases there is clear evidence of a thick disk at the base of the outflow, and it would be quite remarkable if the outflow did not actually start there. Since the disk has no internal source of energy, the power must be supplied from elsewhere, and the radiation from the central star is the only candidate. Therefore, the cyclindrical shape is very suggestive of evaporative outflow.

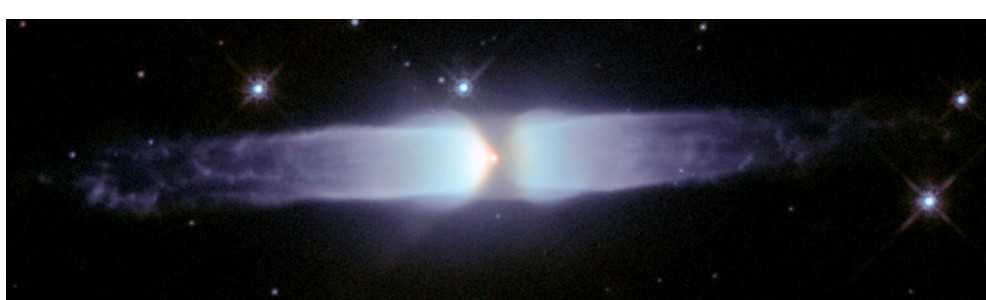

**Figure 1.** Hubble Space Telescope observation of Henize 3-401 [2].

This prompted me to compute a series of hydrodynamical models in which a dense gaseous ring around a hot luminous star is irradiated so fiercely that the disk surface boils off into space. The disk is presumably approximately barotropic, so that its cross section is quite thick; not really a torus, but more akin to a hollow ring. I do not specify an origin for the disk-ring; a plausible possibility is that the central object is a late post-AGB star, on its way to becoming a white dwarf, surrounded by the remnants of its planetary system (or a

super-jupiter binary companion), and/or a gas–dust mixture slowly ejected in an earlier phase of the star's evolution.

## 2. Evaporation Outflow Model

In previous studies [3], I have considered the formation of bipolar nebulae under the assumption that the shaping of the nebula is due to a spherical stellar wind impinging on a disk around the star. I found that this mechanism makes nice bipolars, and that it is possible thereby to explain quite a range of observed nebulae. Others have presented computations based on jet-driven or magnetically confined outflows (e.g., Ref. [4] and references therein).

However, there are many bipolar nebulae that do not show the figure-eight shape typical of such interaction models [5]. In many of these objects the circumstellar disk is quite prominent. Considering this, I decided to look at the launching problem 'the other way around' and assumed that the flow originates directly on the disk, instead of on (or very near to) the star. When a spherical outflow is confined by a disk, it is quite hard to make tall cylindrical outflow shapes, due to the contrast between the Mach number of the stellar wind and that of the circumstellar disk. Therefore, I decided to investigate what the opposite viewpoint offers: namely, that the disk itself is doing the launching.

Suppose that the shape of the ring-shaped disk around the star is given in cylindrical coordinates $(r, \phi, z)$, with a cross section $z(r)$. Consider an infinitesimal annulus with a surface area $\Delta A$ on the ring. Suppose that this annulus is evaporating due to irradiation from the star. Let the velocity $\vec{v}$ be perpendicular to the ring surface, with gas density $\rho$. The amount of energy that flows per second through that surface, with speed $v$, is $\frac{1}{2}\rho v^3 \Delta A$. This energy is delivered by the radiation from the star which has luminosity $\mathcal{L}$. If the surface $\Delta A$ subtends a solid angle $\Delta\Omega$ as seen from the star, then the energy flux that corresponds to the above kinetic energy is $\mathcal{L}\,\Delta\Omega/4\pi$. Equating the two energy fluxes we obtain

$$\frac{1}{2}\rho v^3 \,\Delta A = \frac{\mathcal{L}}{4\pi}\,\Delta\Omega \tag{1}$$

The annulus has an inclination $\psi - \theta$ with respect to the direction towards the star, where $\psi$ is the inclination of the ring surface with respect to the symmetry axis, and $\theta$ is the latitude in spherical coordinates $(R, \phi, \theta)$. Therefore,

$$\frac{\Delta\Omega}{\Delta A} = R^{-2}\sin(\psi - \theta) \tag{2}$$

in which $R = \sqrt{r^2 + z^2}$ is the radial distance from the star to the ring surface (I assume that the radius of the star is negligible). Putting all of this together, we obtain

$$v^3 = \frac{\mathcal{L}}{2\pi\rho R^2}\sin(\psi - \theta) \tag{3}$$

with the stipulation that $v = 0$ when $\psi < \theta$, because then the annulus is in the shadow of the ring itself.

This equation is valid under the assumption that all of the radiative energy is converted into kinetic energy, hereinafter called 'Case A'. However, some of the radiation will be used to heat the surface of the ring. The temperature $T(R)$ of the outflowing gas is determined by the stellar irradiation. I assume that the star and the ring radiate as black bodies, so that the temperature is found from the classical relation $T^4 R^2 = \text{const} \propto \mathcal{L}$. When setting up the outflow boundary conditions, the temperature is not used as such, but enters by way of the gas pressure $P$. Because the gas density is assumed to be constant on the ring surface, the black-body condition implies that $P = P_0\sqrt{R_0/R}$ in which $P_0$ and $R_0$ are fiducial quantities that depend on the local geometry. As an aside, note that this means that the speed of sound on the evaporating surface scales as $s \propto R^{-1/4}$.

The thermal energy density is the gas pressure $P$. Therefore, Equation (3) has to be replaced by

$$(\frac{1}{2}\rho v^2 + P)v = \frac{\mathcal{L}}{4\pi R^2} \sin(\psi - \theta) \tag{4}$$

This description of the launching conditions on the ring will be called 'case B'. In order to prepare this for the numerical computations, I convert the equation to dimensionless form. Each variable $V$ is replaced by $V_0 V$, in which $V_0$ is a fixed fiducial dimensional quantity, and $V$ is now dimensionless. This ploy will show more clearly on what quantities the simulation depends. Writing $v_0 v$, $\rho_0 \rho$, $P_0 P$ and $R_0 R$, I obtain

$$\left(\frac{1}{2}\rho_0 v_0^2 \rho v^2 + P_0 \sqrt{R_0/R}\right) v_0 v = \frac{\mathcal{L}}{4\pi R_0^2 R^2} \sin(\psi - \theta) \tag{5}$$

Factoring out the fiducial quantities gives

$$\left(\frac{1}{2}\rho v^2 + \frac{P_0}{\rho_0 v_0^2}\frac{1}{\sqrt{R}}\right) v = \frac{\mathcal{L}}{4\pi\rho_0 R_0^2 v_0^3}\frac{1}{R^2} \sin(\psi - \theta) \tag{6}$$

This equation may be rewritten in a more transparent form by using

$$s_0^2 = \gamma\frac{P_0}{\rho_0} ; \quad \mathcal{M}_0 = \frac{v_0}{s_0} ; \quad \mathcal{L}_0 \equiv 4\pi\rho_0 R_0^2 v_0^3 \tag{7}$$

in which $s_0$ is the fiducial speed of sound, $\gamma$ the adiabatic index of the gas, $\mathcal{M}_0$ the Mach number, and $\mathcal{L}_0$ the energy flux. Equation (6) then becomes

$$\left(\frac{1}{2}\rho v^2 + \frac{1}{\gamma\mathcal{M}_0^2\sqrt{R}}\right) v = \frac{\mathcal{L}}{\mathcal{L}_0}\frac{1}{R^2} \sin(\psi - \theta) \tag{8}$$

This is an implicit equation for the velocity $v(R)$ in Case B. Using Equation (3) it follows that closer to the star (smaller values of the dimensionless radius $R$) the kinetic energy density $E = \frac{1}{2}\rho v^2$ dominates over the thermal energy density $P$. The dimensionless equation that corresponds to Case A, Equation (3), is

$$\frac{1}{2}\rho v^3 = \frac{\mathcal{L}}{\mathcal{L}_0}\frac{1}{R^2} \sin(\psi - \theta) \tag{9}$$

When setting up the initial conditions, I prescribe the shape of the ring cross section, from which I compute $\psi(R)$ and $\theta(R)$. In Case A, I use Equation (3) to compute $v(R)$ for plausible values of $\rho$, $\mathcal{M}_0$ and $\mathcal{L}/\mathcal{L}_0$. All of these dimensionless quantities are of the order of unity. Numerical experience shows, for example in plots such as Figure 2, that in the inner regions of an evaporating disk the gas pressure does not play a significant role. This is a consolation for the fact that in the present preliminary investigation I have not computed the actual ring structure in any detail. That is why I have used Equation (3) to find the launching speed, and have proceeeded from there.

The initial conditions in the gas that surrounds the ring are even more uncertain. It is often assumed that the gas density obeys the proportionality $\rho \propto R^{-2}$, as is the case in a spherical outflow with constant speed. If the temperature of the gas is due to the radiation from the star, the pressure would behave as it does on the surface of the ring. In that case, the speed of sound would scale as $s \propto \sqrt{P/\rho} \propto R^{3/4}$. I will return to this point in my conclusions.

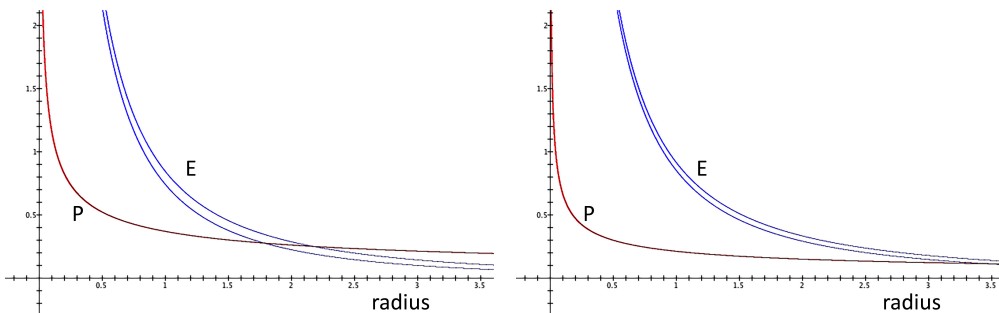

**Figure 2.** Kinetic energy density (blue curves) and gas pressure $P$ (red curves) derived from the implicit form Equation (8) in Case B, for $\mathcal{L}/\mathcal{L}_0 = 1.0$. The left panel has a Mach number $\mathcal{M}_0 = 1.5$, on the right $\mathcal{M}_0 = 2.0$. The values of the adiabatic index are $\gamma = 1$ (isothermal equation of state, lower blue curves) and $\gamma = 5/3$ (adiabatic, upper blue curves).

### 3. Shape of the Ring

Determination of realistic disk-wind launching conditions is a very complicated exercise in radiative transfer. In order to make a start, I have concocted a plausible configuration that I prefer to call a 'ring' rather than a disk, because it is quite thick (as may be seen in the observed images of relevant cases, e.g., Figure 1). Assuming that the configuration is cylindrically symmetric, and taking $(r, z)$ to be the radial and axial cylindrical coordinates, I propose the function

$$z(r) = br^n\{(r_+ - r)(r - r_-)\}^{1/2} \tag{10}$$

in which $r_-$ and $r_+$ are the inner and outer equatorial radii of the ring, respectively; the number $n$ governs the concavity of the ring, typically $n \approx 2$; and $b$ is an arbitrary constant of order unity. This shape may be seen as a plausible approximation to the outer equidensity surface of a barotropic disk.

The surface of the ring is irradiated by the star. The resulting evaporation flow starts with a velocity that is perpendicular to the ring surface, and has a magnitude such that the flux received from the star equals the outflow energy flux (case B) or the outflow energy minus the thermal energy due to the radiative heating (case A). The inclination $\psi(r)$ of the ring and the latitude $\theta(r)$ are determined by

$$\tan\psi = br^{n-1}\{(r_+ - r)(r - r_-)\}^{-1/2}\{-nr_+r_- + (n+1/2)(r_+ + r_-)r - (n+1)r^2\} \tag{11}$$

$$\tan\theta = z/r = br^{n-1}\{(r_+ - r)(r - r_-)\}^{1/2} \tag{12}$$

The speed is found from Equation (3), and because the outflow is assumed to be perpendicular to the ring surface we obtain the launching velocity

$$\vec{v} = v(-\sin\psi, 0, \cos\psi) \tag{13}$$

in cylindrical coordinates $(r, \phi, z)$ (the azimuthal velocity is zero).

In the numerical solutions to be presented below I have adjusted the values of $b$ and $n$, as well as the ratio $r_+/r_-$, until the ring shape looked more or less plausible; that is to say, having a cross section that is sufficiently concave (such that the evaporation velocity is not directed inward too much) but not too thin. Typical shapes and velocity fields are shown in Figure 3. At this stage of the investigation, I have no better argument for adopting these values than noting that $b \approx 1$ and $n \approx 2$ produce cylindrical chimney-type outflows that resemble the observations rather well.

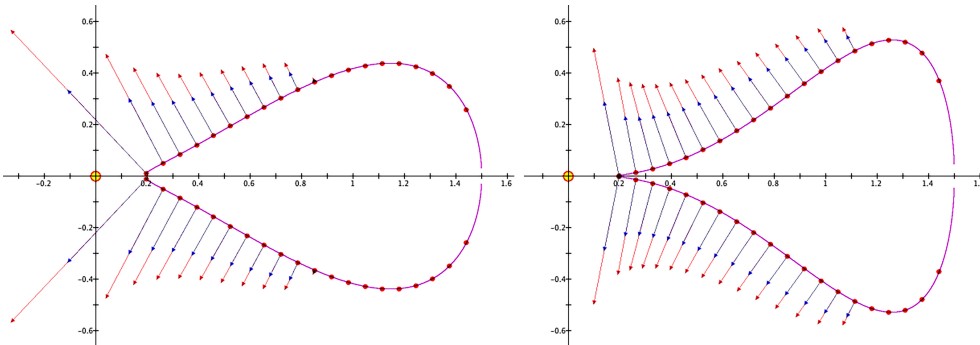

**Figure 3.** Ring model according to Equations (11) and (12) with $r_- = 0.2$, $r_+ = 1.5$, $b = 0.66$, $n = 1$ (**left**) and $n = 2$ (**right**). The blue arrows show the velocity field, the red arrows show the velocities divided by the local sound speed as derived from the pressure in Equation (14) (Mach vectors).

## 4. Hydrodynamical Results

In order to test whether star-driven evaporation produces cylindrical nebulae, I have performed numerical hydrodynamical calculations of outflows driven by evaporation of a pseudo-barotropic ring around a hot central star. My hydrodynamical codes have been presented in [6]. Further improvements that I have made since then do not change the key features of my FCTLCD2D hydro solver, and are almost all concerned with upgrading data handling and computing speed (notably vectorization and parallel loop execution). These are too straightforward to merit separate publication.

To obtain quick results that may indicate whether the chosen approach is fruitful, I have implemented the above initial conditions in my hydrocode FCTLCD2D. I have simplified the setup to a constant gas pressure instead of the dependence in Equation (4). The gas surrounding the ring is assumed to be homogeneous, as opposed to the dependence $\rho \propto R^{-2}$ mentioned in Section 2. The boundaries of the computational grid were fitted with reflectionless outflow conditions.

The images below show a very small sample of the resulting output. All of the results display a pronounced cylindrical outflow pattern. Many of these show a striking similarity to the shape of Hen 3-401 and its kin (Figure 4). It came as a very welcome surprise that even the first runs produced shapes that match the observations rather well. In all the images the [RGB] colour coding is what I habitually use in my work: R = gas density $\rho$, G = pressure $P$, B = speed $v$. In some cases I have increased the contrast and range of the lower values, to make the details of the outflow more prominent.

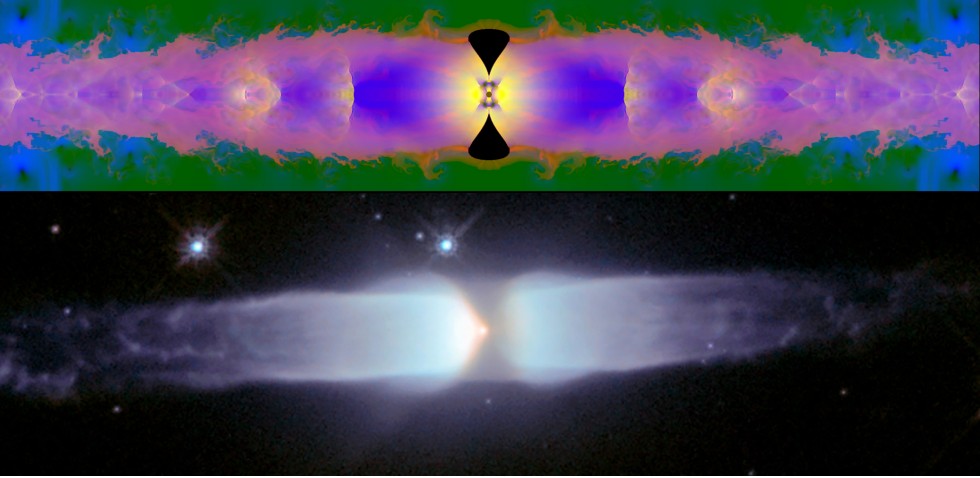

**Figure 4.** (**Top**): cross section snapshot at dimensionless time $t = 6.0$ of the simulation of an evaporating ring as in Equations (11) and (12) with $r_- = 0.15$, $r_+ = 1.0$, $b = 1.5$, $n = 2.5$, $\gamma = 1.2$, $\mathcal{M}_0 = 1.55$ and $\mathcal{L}/\mathcal{L}_0 = 8.0$. RGB colours correspond to $\rho$, $P$, and $v$. (**Bottom**): observation of the cylindrical nebula Hen 3-401 [2].

In all of my simulations, the internal configuration of sequential X-shocks on the axis is beautifully steady, just as in the case of the classical rocket exhaust nozzle. Under astrophysical circumstances, the surrounding gas (the stellar wind deposited before the evaporation flow started) is much less dense than that of the outflow, whereas in the laboratory setup the atmospheric pressure and density are comparatively high. Therefore, the outer walls of the evaporation flow show vortex structures where the outflow mixes with the ambient gas. Even though the shapes of my outflows seem to be suitably similar to what is observed, there is much more involved than just shape.

A typical example of the outflow velocity field is shown in Figure 5. The turbulent eddies on the periphery should be observable. The streaming velocity in the axial direction (middle panel) appears to be constant after the initial acceleration away from the ring. It would seem, however, that in many cases observations show precisely the opposite, namely, a velocity that *increases* approximately linearly with distance from the star. Unless there is an extra source of acceleration (such as the stellar radiation driving the gas after it has left the ring surface), I am afraid that these observations are in tension with the present model. Other flow features that should be compared with observations are identified in Figure 6.

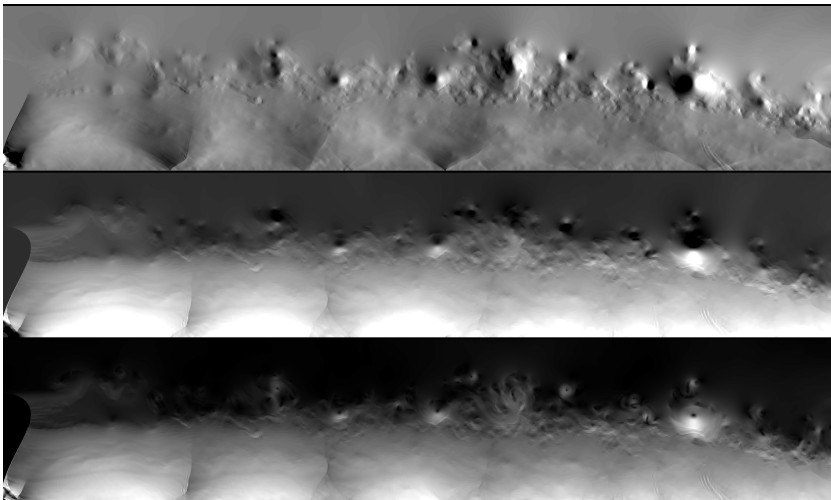

**Figure 5.** Cross section snapshot of the velocities in the simulation of Figure 4. The **bottom** of each panel is the symmetry axis, the ring plane is on the left, and the average direction of the flow is from left to right. (**Top**)—radial velocity, (**Middle**)—axial velocity, (**Bottom**)—total speed. (**Top**): the white shading indicates that the flow is radially outward, whereas the black means that the flow is inward, showing an irregular vortex street. (**Middle**): lighter colours mean higher velocity away from the evaporating ring, whereas darker colours indicate velocity in the opposite direction.

After completion of this series, I have investigated the consequences of the difference between Equation (8) (Case B) and Equation (9) (Case A) for the shaping of the nebula. In Case B, I assume that the temperature, and therefore the pressure, of the evaporating surface is set by the irradiation from the star. Therefore, the energy available is divided between kinetic and thermal energy. In the latter case I assume that the pressure is given somehow, for example by the hydrostatic conditions in the ring, so that all of the stellar energy is turned into the kinetic energy of the evaporating flow.

A typical example is shown in Figure 7. The 'kinetic-only' Case A (Equation (9), top panel) produces a cylindrical nebula. The nebula in the 'kinetic-plus-thermal' Case B (Equation (8), bottom panel) is much more complex. The reason for this is that close to the star, the kinetic energy of the outflow always dominates (compare Figure 2); however, near the place where the stellar light grazes the ring, the speed drops to zero, whereas the pressure does not.

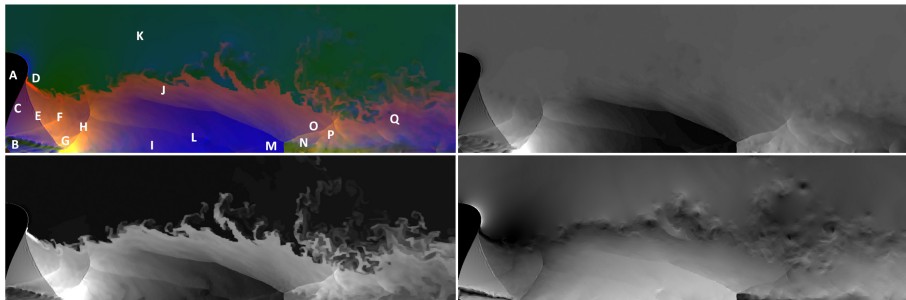

**Figure 6.** (**Top left**): the main features of the evaporating-ring flow pattern (RGB encoded). Case B with $r_- = 0.15$, $r_+ = 1.0$, $b = 1.0$, $n = 2.0$, $\gamma = 1.2$, $\mathcal{M}_0 = 1.55$ and $\mathcal{L}/\mathcal{L}_0 = 8.0$. A—sagittal cross section of the ring. B—cavity around the central star. C—supersonic evaporation flow. D—subsonic evaporation sheath. E—entrainment shock where the supersonic flow meets the subsonic sheath. F—shocked outflow. G—high-pressure pocket where the evaporation streams collide. H—secondary shock emanating from the high-pressure pocket. I—postshock expansion flow with weak shocks on the axis. J—shocked gas sheath, K—gas surrounding the ring. L—turbulent peripheral layer. M—Mach disk at the second X-shock. N—turbulent slip layer behind the Mach disk. O—second branch of the axial X-shock in the inward-curved sheath. P—postshock compressed sheath gas. (**Bottom left**)—gas density (R channel); top right—pressure (G channel); bottom right—speed (B channel).

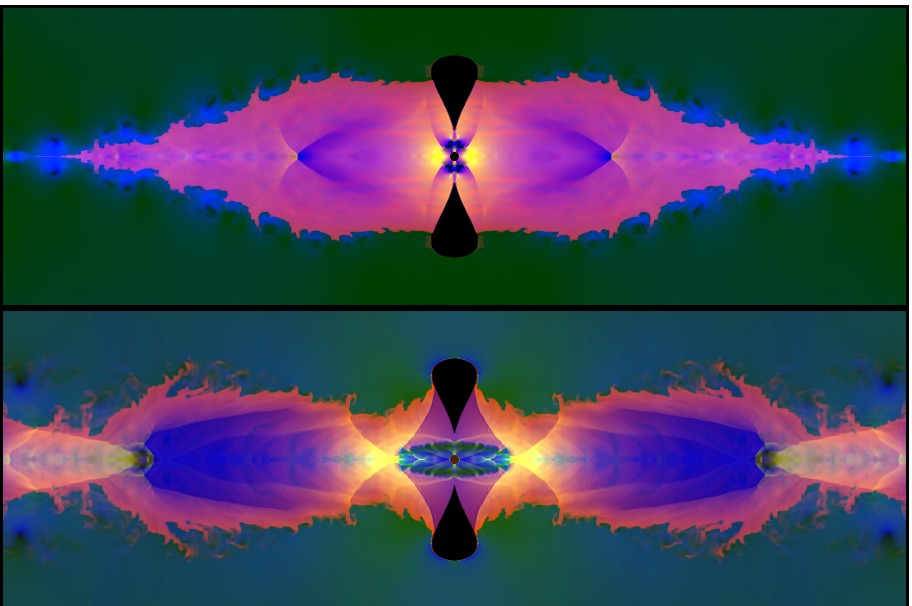

**Figure 7.** Cross section snapshot of two simulations with identical setup for geometry, boundary conditions, and irradiation. (**Top**): Case A initial conditions in Equation (9), so that all of the stellar energy is turned into the kinetic energy of the evaporation. (**Bottom**): Case B initial conditions in Equation (8), wherein the energy available is divided between kinetic and thermal energy.

Consequently, the place near the terminator on the ring (beyond which the star cannot be seen) always produces a dense slow outflow. Further inward, this gas meets the evaporating flow closer to the star and is dragged away from the ring. That interaction makes the flow that intersects on the axis very complicated. Further out, this entrained gas stays in the outer shell of the nebula, leaving a fast low-density stream near the axis. Likewise, the secondary shock that arises when the inward curving shell bends back towards the axis is very different from the more or less ordinary diamond-shock of the case described by Equation (9) (cf. Figures 4 and 5).

The influence of the ring concavity on the outflow conditions may be seen in Figure 3. The hollower the ring, the more radiation is intercepted by the outer regions. Closer to the

star, the light impinges in a more grazing direction, wherefore it is effectively less intense; on the other hand, the surface curvature focuses the initial outflow more in the direction of the symmetry axis. I have investigated what consequences the ring concavity has for the resulting flow pattern. Some results are shown in Figure 8. A bowl-shaped ring produces the widest outflow cylinder, whereas the ring with a conical inner surface makes the flow narrower and almost jet-like.

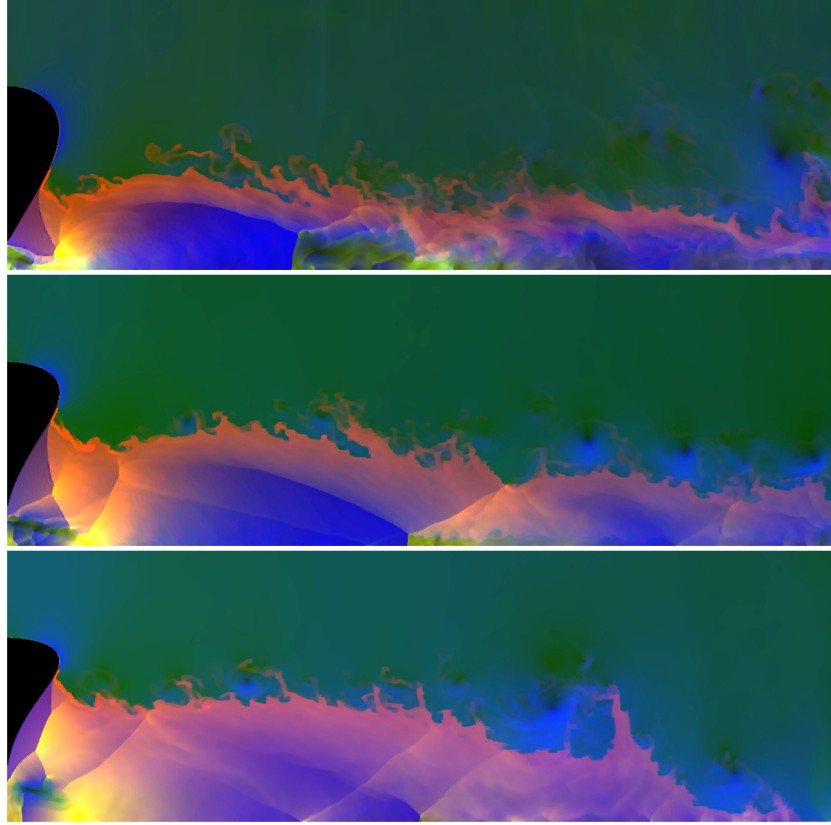

**Figure 8.** Dependence of the flow pattern on the shape of the ring. Colour coding as before: R = gas density $\rho$; G = pressure $P$; B = speed $v$. All simulations Case B; $\mathcal{L}/\mathcal{L}_l = 4$; $r_- = 0.15$; $r_+ = 1.0$; $\gamma = 1.2$; $\mathcal{M}_0 = 1.5$; snapshot at time $t = 1.6$. (**Top**): $n = 1$; $b = 1.0$. (**Middle**): $n = 2$; $b = 1.25$. (**Bottom**): $n = 3$; $b = 1.44$. The shock patterns are globally similar in all cases. The flow that starts from a bowl-shaped ring (**Bottom**) is broadest, whereas the ring with a conical inner surface (**Top**) produces a narrower and better collimated outflow.

## 5. Application to Henize 3-401

To connect with the available observations, I have to map the computed quantities onto the properties that are observed in actual nebulae. A case in point is Henize 3-401, because the shape of that nebula is very suggestive of evaporative outflow. The images show no indication of interaction between the disk and a stellar wind, or a jet. A stellar wind, confined by the disk, would produce a bipolar nebula, but with the shape of the classical Figure 8, and not the elongated cylinder observed in Hen 3-401 and similar morphologies. A jet would be preceded by a bow shock, which is not observed.

I have assembled the most recent values (as far as I can determine) of the observed properties of Hen 3-401 from [7–10]. Unless indicated otherwise, the calculations below all refer to the data listed in Table 1.

**Table 1.** Summary of observed data of Hen 3-401.

| Variable | Quantity | Note |
|---|---:|---|
| length/width ratio | 1:7 | whole nebula |
| lateral expansion | 15.5 km/s | |
| terminal lobe speed | 200 km/s | |
| CO line core speed | 22 km/s | |
| $H_\alpha$ wings out to | 1600 km/s | |
| inclination | 10–15° | w.r.t. sky plane |
| lobe age | 1100 yr | approximately |
| central star | B1e | |
| luminosity B1V | $1.35 \times 10^4$ $L_\odot$ | |
| temperature B1V | 26,000 K | |
| luminosity nebula | $3.6 \times 10^3$ $L_\odot$ | 27% $L_*$? |
| distance | 3 kpc | |
| lobe width | $1.11 \times 10^{15}$ m | 7400 AU |
| lobe radius | $5.6 \times 10^{14}$ m | 0.5 width |
| lobe length | $8 \times 10^{15}$ m | 7 × width |
| momentum | $9 \times 10^{37}$ g cm s$^{-1}$ | |
| kinetic energy | $3 \times 10^{44}$ erg | $3 \times 10^{37}$ J |
| mass | 0.01 $M_\odot$ | $1.99 \times 10^{28}$ kg |
| momentum | $6000\, \mathcal{L}/c$ | |

## 6. Gauging of Dimensionless Variables

My computations of evaporation flows from circumstellar rings were performed in dimensionless units. Each variable $a$ is scaled by writing $A = a/a_0$, where $a_0$ is a fiducial quantity to be determined for each specific case. The equations to be solved are cast in terms of the resulting dimensionless variables $A$. The primary scaling quantities are the mass density $\rho_0$, disk radius $R_0$, flow speed $v_0$ and luminosity $\mathcal{L}_0$. These must be related, because the only independent quantities are mass, length and time. Using Equation (7), we may compute the value of the mechanical luminosity $\mathcal{L}_0$ for some appropriate values of the fiducial quantities $\rho_0$, $R_0$ and $v_0$.

Unless the distance to Hen 3-401 has been grossly overestimated, we may consider $R_0$ as fairly safe, but it is not easy to assess the mass density $\rho_0$. The data from [7] suggest $n_H = 3 \times 10^{15}$ m$^{-3}$. Another way of estimating $\rho_0$ is by assuming that the nebular mass $0.01\, M_\odot = 1.99 \times 10^{28}$ kg is distributed uniformly in a cylinder with the size of the nebula. For the above value of $R_0$, the volume is $\pi R_0^2 H = 1.9 \times 10^{46}$ m$^3$, in which $H \approx 28\, R_0$ is the total length of the nebula. In that case, $\rho_0 = 10^{-18}$ kg m$^{-3}$, corresponding more or less to the lowest value reported [7], namely $10^9$ m$^{-3}$, that is $\rho_0 = 2 \times 10^{-18}$ kg m$^{-3}$. With these numbers, we obtain a particle density of $6.3 \times 10^8$ m$^{-3}$ and $\mathcal{L}_0 = 4.5 \times 10^{27}$ W $= 12\, L_\odot$.

Browsing through the observations, it seems clear that the determination of the length scale $R_0$ is probably the most robust. The speed $v_0$ is next, although its value varies a lot in space, and the inclination of the axis of Hen 3-401 with respect to the plane of the sky is somewhat uncertain. Because $v_0$ appears in the expression with a third power, a relatively small adjustment would produce a noticeable change in $\mathcal{L}_0$. However, $\rho_0$ is the least well known, and it seems reasonable to compute $\mathcal{L}_0$ for a range of densities to see if there are values of $\rho_0$ that yield acceptable values of the mechanical luminosity. The result is that densities in the neighbourhood of $10^{-15}$ kg m$^{-3}$ (or $6 \times 10^{11}$ m$^{-3}$) produce mechanical luminosities that more or less correspond to what the central star of Hen 3-401 might deliver. This falls comfortably in the range [7] from $10^9$ m$^{-3}$ up to $10^{15}$ m$^{-3}$.

There is one more constraint on the mass density $\rho_0$ to be considered. Because the outflow is presumably driven by the irradiation from the central star, the density must be such that the distance $R_0$ corresponds at least to an optical depth of approximately unity. This places an upper limit on the mass density. That is to say, if the mean scattering cross section of the nebular matter is $\sigma$, then $R_0 \approx 1/n_0\sigma$ so that $\rho_0 \approx m/\sigma R_0$ where $n_0$ is the particle density corresponding to $\rho_0$, and $m$ is the mean particle mass. For example,

using the Thomson cross section $\sigma_T$, and taking $R_0 = 6 \times 10^{14}$ m from the above, we obtain $\rho_0 = m_H / \sigma_T R_0 = 4.2 \times 10^{-14}$ kg m$^{-3}$ and $n_0 = 2.5 \times 10^{13}$ m$^{-3}$. This upper limit is fairly consistent with the data in Table 1. All of this considered, I think that $\rho_0 = 10^{-15}$ kg m$^{-3} = 6 \times 10^{11}$ m$^{-3}$ and $\mathcal{L}_0 = 4 \times 10^{30}$ W $= 10^4$ L$_\odot$ are plausible values.

One more parameter is essential in the hydro simulations, namely the initial Mach number $\mathcal{M}$ of the evaporating outflow. Because the circumstellar ring is supposed to have no energy sources other than the stellar irradiation, we may assume that $\mathcal{M} \approx 1$, so that $v_0$ is approximately equal to the local sound speed $s$. To determine $s$, we must know the local gas pressure $P$, which may be derived from the local temperature $T$ by means of the ideal gas law:

$$P = \frac{k}{m} \rho T ; \quad s^2 = \gamma \frac{P}{\rho} = \gamma \frac{k}{m} T ; \quad L = \sigma A T^4 = 4\pi\sigma R^2 T^4 \tag{14}$$

where $k$ is Boltzmann's constant, $m$ is the mean particle mass, $A$ is the stellar surface, and $\sigma$ is the Stefan-Boltzmann constant. In my evaporation model, $T_0$ is due to the central star. If the central star is B1, then $T = 2.6 \times 10^4$ K and $L = 1.35 \times 10^4$ L$_\odot$ $= 5.2 \times 10^{30}$ W, and using $r = 10^{14}$ m we obtain

$$T = \left( \frac{L}{4\pi\sigma R^2} \right)^{1/4} = 164 \text{ K} \tag{15}$$

which compares fairly well with the IRAS values [10]. Taking the IRAS temperature they report, 350 K, I obtain $s = \sqrt{\gamma k T / m_H} = 2.2$ km/s which should be seen as a lower limit, because the gas temperature is expected to be larger than the dust temperature.

Finally, there is one more quantity that should be compared with the local speed of sound, namely the orbital speed $v_d$ of the disk at radius $R_0$: $v_d = \sqrt{GM_*/R_0} = 4$ km/s if $M_* = 11$ M$_\odot$ and $r_0 = 10^{14}$ m (B1 star). If the disk is in hydrostatic equilibrium, the local sound speed should be about an order of magnitude less. We may conclude that the Mach number of the initial outflow would be in the range 1–5. Most of the simulations I have run have a Mach number in this range.

## 7. Discussion and Conclusions

My original aim was a preliminary investigation to find out if a circumstellar ring, irradiated by the central star, would produce a cylindrically collimated outflow. The conclusion is that it does, for a plausible family of ring shapes, and under quite a wide variety of launching conditions. The generic flow features, described in Figure 6, are very suggestive of features found in many pseudocylindrical bipolar nebulae, as far as the observational resolution allows. However, a morphological match is merely a minimum requirement of any theoretical model. This should be followed up by the construction of observables, which I will reserve for a future publication.

Two more avenues for future research are: the inclusion of rotation about the symmetry axis, and fully three-dimensional hydrodynamics. The azimuthal speed estimated above might not be very much smaller than the axial speed on launch. It is to be expected that the conservation of angular momentum would prevent the flow from focusing too strongly on the axis, and might contribute to the increase of the density on the cylinder walls. Simulations of rotating flows are readily possible in two dimensions, and I will work on that in the future. The 3D avenue is more technically demanding and will have to wait.

The main item of concern is the velocity field. In the majority of the computational runs, the outflow is only weakly accelerated. This is at variance with the observation that in many such nebulae, the speed of the outer walls increases approximately linearly with the distance along the cylinder. This shortcoming may be due to my initial choice for the distribution of the gas surrounding the ring, which I have given a constant and rather low mass density. As mentioned in Section 2, the gas density in a spherical stellar atmosphere expanding at constant speed obeys $\rho \propto R^{-2}$. If the temperature of the atmosphere is due to the radiation from the star, the speed of sound would scale as $s \propto \sqrt{P/\rho} \propto R^{3/4}$, so

that the Mach number would decrease away from the star: $\mathcal{M} \propto R^{-3/4}$. If the density of the gas into which the outflow propagates is much larger, and if indeed $\rho \propto R^{-2}$, the outer walls would be much denser, and would couple to the ambient gas through turbulent viscosity (cf. the eddies in Figure 5). These effects together might produce an acceleration with $v \propto z$, in accordance with the observations (see also [11]). To make matters even more complicated, one should bear in mind that the late stages of stellar evolution on the red giant branch produce outflows that become faster in the course of time, and may even be episodic. A possible consequence, proposed by an anonymous referee, is that a secular decrease of the stellar luminosity leads to a decrease of the launching speed.

Published observations do not readily provide the main physical quantities needed for theoretical modeling. In particular, it is not clear to what part of the nebula the various quantities refer. Moreover, the error margins are difficult to assess. I am not qualified to judge the quantitative aspects of the observations, but purely from the nebular shapes I would say that, for example, Hen 3-401, He2-437, OH231.8 and V445Pup are fitted very well by my models, without undue effort in adjusting the outflow parameters.

My model has at least this virtue, that the source of the driving energy (the light from the central star) and the cause of the collimation (the hollow inner surface of the ring) are evident (as opposed to, e.g., jet-driven models, where the mechanism that causes the jet in the first place is often unspecified). This is obviously not enough to be an incontrovertible explanation for nebulae of the Hen 3-401 type, but it is a start.

**Funding:** This research was supported by Universiteit Leiden, under the usual conditions for University employees.

**Institutional Review Board Statement:** Not applicable.

**Informed Consent Statement:** Not applicable.

**Data Availability Statement:** Not applicable.

**Acknowledgments:** I am grateful to Bruce Balick for his enthusiasm and support, and for drawing my attention to cylindrical nebulae as exemplified by Hen 3-401. I thank the referees for their prompt, perspicacious and helpful remarks and corrections.

**Conflicts of Interest:** The author declares no conflict of interest.

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
