# Peer review of "Cylindrical Planetary Nebulae. I. Flow from an Irradiated Ring"

_galaxies, doi:10.3390/galaxies10020053_

Round 1

Reviewer 1 Report

The paper presents a very interesting exploration of pre-PNe produced through the photoevaporation of a ring-like circumstellar structure. This work is original, and might lead to substantial future work on this possibility.

I have a number of small comments, which I include here (these are mostly language corrections):

abstract

no indication whatever -> no indication whatsoever

But there is -> However, there is

sec. 1

the first paragraph uncannily resembles the abstract. Is this ok?

no indication whatever -> no indication whatsoever

possibility is, that -> possibility is that

sec. 2

the choice of "R" for the cylindrical radius, and "r" for
the spherical radius is somewhat odd! (the opposite notation choice
is generally made)

sec. 5

no indication whatever -> no indication whatsoever

sec. 7

I do not understand the argument about how to get v \propto z. If
the flow leaves the surface of the ring sonically, one would expect
to get at most an acceleration of a factor of $\sim 2$ (from Bernoulli's
theorem), and not an acceleration to much higher velocities at larger
distances. What about the possibility of the wind starting at
larger disk radii as a function of increasing times? could this
produce a "deceleration" of decreasing outflow velocities vs. time?

Author Response

Leiden, 10-03-2022

Dear Colleague,

Thank you for your prompt and relevant comments and criticisms. I agree with all of them, and have amended my manuscript accordingly. In particular, I am grateful for your suggestion regarding the initial velocity field (Sect.7).

Best regards,

Vincent Icke

===========

Comments and Suggestions for Authors

The paper presents a very interesting exploration of pre-PNe produced through the photoevaporation of a ring-like circumstellar structure. This work is original, and might lead to substantial future work on this possibility.

I have a number of small comments, which I include here (these are mostly language corrections):

abstract

no indication whatever -> no indication whatsoever

But there is -> However, there is

sec. 1

the first paragraph uncannily resembles the abstract. Is this ok?

no indication whatever -> no indication whatsoever

possibility is, that -> possibility is that

sec. 2

the choice of "R" for the cylindrical radius, and "r" for
the spherical radius is somewhat odd! (the opposite notation choice
is generally made)

sec. 5

no indication whatever -> no indication whatsoever

sec. 7

I do not understand the argument about how to get v \propto z. If
the flow leaves the surface of the ring sonically, one would expect
to get at most an acceleration of a factor of $\sim 2$ (from Bernoulli's
theorem), and not an acceleration to much higher velocities at larger
distances. What about the possibility of the wind starting at
larger disk radii as a function of increasing times? could this
produce a "deceleration" of decreasing outflow velocities vs. time?

Submission Date
11 February 2022
Date of this review
25 Feb 2022 22:24:48

Reviewer 2 Report

-typo in line 20

-Theta in Eq 2 is not defined. (Theta is the azimuthal angle in the coordinate system but  from the description I could not see why that would be in the equation.)

-In line 259, the IRAS temperature is a dust temperature while the discussion is about the gas temperature. There is no obvious reason why these two temperatures should be the same under these conditions

Author Response

Leiden, 10-03-2022

Dear Colleague,

Thank you for your prompt and relevant comments and criticisms. I agree with all of them, and have amended my manuscript accordingly.

Best regards,

Vincent Icke

===========

Comments and Suggestions for Authors

-typo in line 20
-Theta in Eq 2 is not defined. (Theta is the azimuthal angle in the coordinate system but  from the description I could not see why that would be in the equation.)

-In line 259, the IRAS temperature is a dust temperature while the discussion is about the gas temperature. There is no obvious reason why these two temperatures should be the same under these conditions

Submission Date
11 February 2022
Date of this review
26 Feb 2022 05:12:37

Reviewer 3 Report

This paper presents a new model for the formation of cylindrical Planetary nebulae considering the evaporation of an irradiated ring. Two scenarios are explored a) that all the radiative energy is converted into kinetic energy (‘kinetic-only’) and b) that the radiative energy is converted into kinetic energy  and thermal energy ‘kinetic-plus-thermal’.

Preliminary results from this new model obtained from hydrodynamic simulations are indeed very promising. The characteristics as well as the structure of the ring used for the simulations are well explained.

I recommend the paper for publication and congratulate the author for this very interesting idea.

I have just a minor comment.

The term "atmosphere" was quite perplexing to me.

e.g. line 99, it is written "The initial conditions in the atmosphere that surrounds the ring are even more uncertain"

e.g. Figure 6, caption "... K: exterior atmosphere

"e.g. line 99, it is written "...for the outer atmosphere..."

The term atmosphere is usually associated with stars (or planets) and various readers with no experience on PNe or in general extended ionized nebulae may get confused. My suggestion is something like " circumstellar envelope" or "AGB envelope".

Typo:

line 287 -> my intial choice -> initial

Author Response

Leiden, 10-03-2022

Dear Colleague,

Thank you for your prompt and relevant comments and criticisms. I agree with all of them, and have amended my manuscript accordingly.

Best regards,

Vincent Icke

===========

Comments and Suggestions for Authors

This paper presents a new model for the formation of cylindrical Planetary nebulae considering the evaporation of an irradiated ring. Two scenarios are explored a) that all the radiative energy is converted into kinetic energy (‘kinetic-only’) and b) that the radiative energy is converted into kinetic energy  and thermal energy ‘kinetic-plus-thermal’.

Preliminary results from this new model obtained from hydrodynamic simulations are indeed very promising. The characteristics as well as the structure of the ring used for the simulations are well explained.

I recommend the paper for publication and congratulate the author for this very interesting idea.

I have just a minor comment.

The term "atmosphere" was quite perplexing to me.

e.g. line 99, it is written "The initial conditions in the atmosphere that surrounds the ring are even more uncertain"

e.g. Figure 6, caption "... K: exterior atmosphere

"e.g. line 99, it is written "...for the outer atmosphere..."

The term atmosphere is usually associated with stars (or planets) and various readers with no experience on PNe or in general extended ionized nebulae may get confused. My suggestion is something like " circumstellar envelope" or "AGB envelope".

Typo:

line 287 -> my intial choice -> initial

Submission Date
11 February 2022
Date of this review
18 Feb 2022 18:00:39